# Prevalence, Attitudes, and Factors Influencing Uptake of the COVID-19 Vaccine in Saudi Arabia

**DOI:** 10.3390/healthcare11070999

**Published:** 2023-03-31

**Authors:** Ohood Felemban, Ahlam Al-Zahrani, Abdalkarem Alsharari

**Affiliations:** 1College of Nursing, King Abdulaziz University, Jeddah 21589, Saudi Arabia; 2College of Nursing, Jouf University, Sakakah 72388, Saudi Arabia

**Keywords:** COVID-19 vaccines, attitude of health personnel, vaccination acceptance, healthcare surveys

## Abstract

Background: The availability and access to COVID-19 vaccinations are critical to a successful pandemic response. More than 70% of the population reportedly needs to be vaccinated against COVID-19 to achieve herd immunity worldwide. However, the reluctance to get vaccinated with the COVID-19 vaccines is holding up the process of vaccination and efforts to control the pandemic and its negative consequences for the global health system, society, and economy. Previous studies have shown low uptake of vaccination in some Middle Eastern countries due to negative attitudes toward vaccination, including concerns about safety and efficacy and doubts about the need for vaccination. Aim: The aim of this study is to investigate the prevalence, attitudes, and factors influencing COVID-19 vaccine acceptance among healthcare workers, academic staff, and students in Saudi Arabia after the vaccine was made widely available by the government. Method: A cross-sectional survey was conducted to determine the prevalence, attitudes, and association between demographic factors and uptake of the first or second dose of vaccination among Saudi Arabian health workers and students. Data were collected using an online questionnaire administered and distributed through the Qualtrics platform. Results: The study recruited 173 participants from different countries and from different Saudi regions, most of whom were faculty members (n = 83). Results indicated significant differences between regions; the mean attitude score for the Western region (M 3.23) was significantly higher than that for other regions (M 3.08, *p* = 0.030). There was also an association between education level and number of vaccine doses received. Thus, the participants with higher education were the most compliant with national vaccination requirements (*p* = 0.004). Although the three professional groups reported social media as the most frequently reported source of information (*p* = 0.021), administrators were more likely to receive information from the MOH than other professional groups. Similarly, faculty members were more likely to receive information from colleagues and professional journals than the other two professional groups. Conclusions: Government officials should build public confidence through vaccination campaigns and devise effective health education programs to increase vaccination uptake. Authorized institutions can effectively use social media platforms to encourage vaccination and promote awareness among all audiences.

## 1. Introduction

The first case of coronavirus disease (COVID-19) was detected in Wuhan, China, in December 2019. Soon after, it became a global pandemic and public health threat [1]. Morbidity and mortality rates from COVID-19 are increasing daily. According to the World Health Organization (WHO) Coronavirus COVID-19 Dashboard [2], there have been 151,803,822 confirmed cases and 3,186,538 deaths as of 2 May 2021. The COVID-19 pandemic has high morbidity and mortality rates that will negatively impact societies and economies worldwide if not brought under control [3]. The virus is highly contagious and spreads by direct and contact transmission [4]. Since the beginning of the pandemic, people have relied on various preventive measures such as social distancing, hand hygiene, and wearing masks. Accordingly, researchers around the world have conducted numerous trials to develop an effective vaccine to combat the pandemic [5]. A few countries have succeeded in producing a COVID-19 vaccine, including the United States (U.S.), China, Russia, and the United Kingdom [6]. In addition, several regions have been able to identify community-specific needs, recognize self-reliance, coordinate public health policies, and develop a sustainable, comprehensive healthcare approach to overcome the COVID-19 pandemic [7].

A high COVID-19 vaccination coverage rate is essential for a successful pandemic response. Research estimates that more than 70% of the population will need to be vaccinated against COVID-19 to achieve herd immunity worldwide [6,7,8]. However, reluctance to get vaccinated with the COVID-19 vaccine is one of the main problems facing public health organizations worldwide as they try to control the pandemic and limit its undesirable effects on health, society, and economies around the world [5,6]. In 2015, the WHO defined vaccine delay as a delay in accepting or refusing immunizations despite the availability of immunization services [2]. According to WHO, vaccine hesitancy has been ranked as one of the top ten global health threats in 2019 in several countries. This poses a major challenge to achieving immunity in the community and raises pandemic concerns worldwide.

Vaccination acceptance is the decision of an individual or group to agree or disagree with a vaccination when the opportunity to be vaccinated presents itself [9]. It might also be considered as accepting COVID-19 as a real threat not only the value of vaccination [10]. We felt that healthcare professionals were knowledgeable about drugs and their associated effects on subjects were likely to reject the COVID-19 vaccine because of its accelerated development phase and lack of clear information about its efficacy and safety. Literary evidence has shown that the low uptake of vaccination may be related to people’s doubts about the need for vaccination, apart from their concerns about the safety and efficacy of the vaccine [5,6,7,8,9,10,11,12]. People’s knowledge, attitudes, and beliefs about COVID-19 vaccination and barriers to vaccination are also a reason for hesitancy [13,14]. The most common reasons for not accepting the COVID-19 vaccine included concerns about adverse health effects after vaccination and the acceptance of information disseminated via social media [15]. It is necessary to study these factors to address the problem of vaccine hesitancy in the population to increase vaccine acceptance, and increase the levels of COVID-19 immunity in the population [11,12,13,14,15,16]. Since the beginning of the pandemic, several studies have examined COVID-19 vaccine acceptance in a specific population group, as well as knowledge about the vaccine and people’s attitudes and beliefs toward the vaccine [5,6,7,8,9,10,11,12,13,14,15,16].

Studies were conducted to evaluate the acceptability of the COVID-19 vaccines [12,13,14,15,16,17,18,19,20,21]. A systematic review was conducted to provide an up-to-date assessment of the acceptability of the COVID-19 vaccines worldwide, and it included results from 33 countries [17]. It was found that COVID-19 vaccine acceptance is low in several regions, including the Middle East. Notably, the lowest COVID-19 vaccine acceptance was found in Kuwait (23.6%), followed by Jordan (28.4%) and Saudi Arabia (64.7%). Another study examined COVID-19 vaccine acceptance in Kuwait and reported low acceptance (53.1%). [21].

Several studies have been conducted to determine the associated factors [18,19,20,21,22,23]. Studies have shown an insufficient willingness to be vaccinated with the COVID-19 vaccine and have identified several factors associated with this hesitancy. These include demographic factors such as age, education level, gender, income, residence, occupation, marital status, race/ethnicity; people’s concerns such as the perceived risk of COVID-19 [20,21]. Trust in the health care system, health insurance, vaccination recommendations, insufficient data on the adverse effects of the vaccine, perceived effectiveness of the COVID-19 vaccine, concerns about the safety of the vaccine, personal factors such as testing for COVID-19 in the past, attitude toward the vaccine, vaccination history, presence of a chronic illness, and working in health care [17,18,19,20,21,22,23,24,25].

Studies have also examined factors that increase vaccine acceptance and influence people’s decision to get vaccinated [1,2,3,4,5,6,7,8,9,10,11,12,13,14,15,16,17,18,19,20,21,22,23,24,25,26,27,28]. Factors include the availability of the vaccine at zero cost [1,2,3,4,5,6,7,8,9,10,11,12,13,14,15,16,17,18,19,20,21,22,23,24,25,26,27], positive beliefs and attitudes about COVID-19 vaccination [22,23], confirmation of the safety and efficacy of the COVID-19 vaccine [22], compulsory vaccination by the government or employers [22], or if recommended by physicians [22,23,24,25,26,27]. Other factors could include a higher level of education, working in government institutions, high trust in the health care system, lack of previous COVID-19 infection experience, possession of the latest COVID-19 vaccine information, being a male subject, access to shared information through government websites and trusted news outlets, vaccines being available in multiple locations, high levels of health literacy, and a sense of responsibility for containing the current pandemic [1,2,3,4,5,6,7,8,9,10,11,12,13,14,15,16,17,18,19,20,21,22,23,24,25,26,27,28,29].

Despite the aforementioned evidence, the decision to accept or reject COVID-19 vaccines may vary depending on people’s geographic location and culture. Therefore, this study is unique in the sense that it examines the factors that might influence the acceptance and use of COVID-19 vaccines in the Kingdom of Saudi Arabia (KSA). We, therefore, intend to investigate COVID-19 vaccination coverage among healthcare college employees and students, determine their attitudes toward the vaccine, and identify the factors that influence their decision to be vaccinated and their correlates in the KSA. This is because vaccine hesitancy may be the main problem facing public health organizations worldwide as they attempt to control the COVID-19 pandemic or similar pandemics. To achieve herd immunity, it is also necessary to understand the reasons for low vaccine uptake in order to overcome the challenge of vaccine hesitancy through a comprehensive vaccination campaign. Thus, the results of this study might illustrate the methods that governments can use to increase public confidence and willingness to vaccinate.

## 2. Materials and Methods

The study was based on a quantitative cross-sectional survey using a convenient sampling technique to determine perceptions of and attitudes toward the COVID-19 vaccine among staff and students at health colleges in KSA. All participants were able to read English or Arabic and agreed to participate in this study. This allowed us to maximize recruitment and cover a wider geographic area. The data collection period was from April 2021 to April 2022 using a self-report electronic questionnaire and Qualtrics software. Survey items were developed by the researchers and guided by the literature to meet the research objectives.

### 2.1. Measures

The survey consisted of four parts: (a) demographic information, including age, nationality, gender, education level, region, college name, and occupation, (b) COVID-19 vaccination information, including the source of information about COVID-19, availability of a COVID-19 vaccination center at current workplace, previous COVID-19 infection, the severity of symptoms, previous COVID-19 infection of family members, number of COVID-19 vaccine doses received, reasons for non-vaccination COVID-19 and brand name of the vaccine. Section (c) contained ten questions on attitudes toward COVID-19 vaccination. We used a five-point Likert scale ranging from 5—strongly agree to 1—strongly disagree. The questions included (1) I feel there is no such virus, and therefore, no need for receiving the vaccination, (2) I am hesitant to receive the COVID-19 vaccine because it has serious side effects and might lead to hospitalization or death, (3) I do not trust the COVID-19 vaccine benefit, (4) I feel the reasons for promoting the COVID-19 vaccine is for the financial profit of pharmaceutical companies, (5) I feel no need for the vaccination of people who had coronavirus because they are immune, (6) I am very concerned that I might have COVID-19 virus at the time of vaccination, (7) I think receiving the COVID-19 vaccine will increase the immunity in the community, (8) I support receiving COVID-19 vaccine because it will enable to remove all restrictions and going back to normal life pattern, (9) It is important to receive the COVID-19 vaccine to control the spread of the disease, and (10) It is important for me to receive the COVID-19 vaccine to protect others and myself. The higher scores chosen indicated higher levels of positive attitudes toward vaccination. However, because the scale contains a mixture of positive and negative items, we reversed the negative items before analysis. The total score of attitude was calculated as the mean score of ten items. Section (d) contained nine questions on factors influencing the uptake of COVID-19 vaccinations. This includes: (1) fear of the unforeseen future effects of the COVID-19-vaccine would decrease the tendency to receive it; (2) mandating vaccine for Haj and Umra would increase the tendency to receive it; (3) free vaccines would increase the tendency to receive it; (4) Some people prefer natural immunity this may decrease the tendency to receive COVID-19 vaccine; (5) providing education about the COVID-19 vaccine and its benefits and side effects would increase the tendency to receive it; (6) mandating the COVID-19 vaccine by law would increase the tendency to receive it; (7) seeing a family member or friend receiving the COVID-19 vaccine would encourage others to receive it if their experience were positive; (8) people with risk factors tend to receive the COVID-19 vaccine, and (9) the safety and effectiveness of vaccines would encourage people to receive it. Ratings were conducted using a five-point Likert scale ranging from 5—strongly agree to 1—strongly disagree.

### 2.2. Data Analysis

SPSS version 25 (IBM SPSS Statistics for Windows, version 25.0, IBM Corp, Armonk, NY, USA) was used for data analysis. Categorical variables were expressed as frequencies and percentages. Quality control of responses was performed using Qualtrics software. Survey item options were enforced by the system during data collection to obtain the most complete responses possible. However, incomplete surveys (less than 70% of survey items) were subsequently deleted. A five-point Likert scale ranging from 5—strongly agrees to 1—strongly disagree was used for ratings. The higher the selected values, the more positive the attitude toward vaccination. The overall attitude score was calculated as the mean of the ten items. We classify the attitude as either positive or negative based on Bloom’s cut-off point method. The original Bloom’s cut-off points of ≤60% as poor, 60–79% as moderate, and 80–100% as good were adopted and modified [30]. Aiming for the best outcome, we classified attitude as positive if it was >80% and negative if it was <80%. Sociodemographic data were analyzed using frequencies and percentages. Logistic regression was used to examine the association between the independent variables (i.e., age, education level, occupation, region, and nationality) and the dependent variables (attitude scores and uptake of the first or second vaccination dose). Pearson’s chi-square test was used to assess differences between categorical data sets in this study. Statistical significance was defined as a *p*-value < 0.05. First, we examined the association between demographic factors and attitude scores. We then examined the associations between demographic factors and uptake of the first or second dose of vaccination.

## 3. Results

### 3.1. Demographic Data

As shown in Table 1, a total of 173 people participated in the survey. Most participants were Saudi nationals (73.99%); however, 36.99% of the study population had PhDs and 47.98% were faculty members. Most participants were female (75.14%), and 72.25% were from the Western region. Most participants obtained their information about COVID-19 mainly from social media (58.96%). In addition, 87.86% of participants reported that there was a COVID-19 vaccination center at their workplace (Table 1).

### 3.2. Prevalence of COVID-19 Infection

Evidence from this study has shown that 84.97% of the participants had never been infected. Of those who had ever been infected with COVID-19, 80.77% had only mild symptoms. Almost half of the participants (50.29%) reported that none of their family members had been infected with COVID-19; 89.02% of the study population had received COVID-19 vaccines, with 65.58% receiving a second dose. Pfizer appeared to be the most used vaccine (81.16%). When those who received the first dose of the COVID-19 vaccine were asked about the reasons for not receiving the second dose, safety concerns (30.19%), lack of intent (20.75%), and other reasons (33.97%) were reported (Table 2).

### 3.3. Factors Influencing Receipt of COVID-19 Vaccination

Participants reported that the following factors could have a strong influence on vaccination with the COVID-19 vaccine. The majority of the participants agreed strongly that education about the COVID-19 vaccine and its benefits and side effects (M = 4.45), (n = 105, 63.3%); being told that the vaccine is required for Haj and Umra (pilgrimage) visas (M = 4.42), (n = 109, 65.7%); seeing a family member or friend vaccinated with COVID-19 (M = 4.40), (n = 101, 60.8%); vaccine safety and efficacy (M = 4.39), (n = 107, 64.5%); free vaccines (M = 4.36), (n = 102, 61.4%) and being required by law to be vaccinated with COVID-19 (M = 4.20), (n = 96, 57.8%) (Table 3).

### 3.4. Attitude toward COVID-19 Vaccination

A significant relationship was found between male and female participants. The difference in attitude scores was found between demographic variables. Specifically, the mean total attitude score was significantly higher in the West region (M = 3.23, SD = 0.40) than in the other regions (M = 3.08, SD = 0.44), *p* = 0.030. However, the results for age were not significant. Of all demographic factors, only education was significant Specifically, holders of master’s and doctoral degrees are most likely to comply with the vaccination requirement, while the holders of bachelor’s degrees are less likely to receive the required two doses of vaccination (0.05, χ^2^ (4) = 15.33, *p* = 0.004) (Table 4).

We also found a relationship between the participants’ occupation and their main source of information COVID-19 (0.05, *p* = 0.021). Although the three professional groups reported social media as the most frequently reported source of information (*p* = 0.021), administrators were more likely to receive information from the MOH than other professional groups. Similarly, faculty members were more likely to receive information from colleagues and journals than the other two professional groups (Table 5).

Finally, the perceived benefits that the COVID-19 vaccine could increase immunity (n = 104, 60.1%), and the possible control of the spread of the disease (n = 128, 74.0%), the possible removal of limitations leading to a return to normal life (n = 109, 63.0%), and the desire for self-protection (n = 132, 76.3%) are likely some of the factors influencing their attitudes (Table 6).

## 4. Discussion

It is our understanding that this study provides evidence of perceived factors influencing COVID-19 vaccine acceptance in the Kingdom of Saudi Arabia. Professional affiliation with health care might contribute to vaccine acceptance in this cohort. Therefore, exploring people’s knowledge of this disease, their attitudes toward it, and their intentions to get vaccinated is critical to curbing the spread of COVID-19, protecting communities, and promoting the health of the future generation.

The results of this study showed that the number of participants who received the COVID-19 vaccine was relatively high (89.02%). This result supports the study [15] which found that COVID-19 vaccine acceptance in Saudi Arabia was 64.7%. In addition, a study found that the acceptance of vaccination and the intention to be vaccinated was 79.2% in the Saudi population [28].

Compared to other countries, we found a high uptake of COVID-19 vaccinations in Saudi Arabia. This result may be due to the tremendous efforts of the Saudi government to raise awareness of the vaccine by offering COVID-19 vaccination free of charge and mandating COVID-19 vaccination in the country with strict monitoring and follow-up policies, such as before entering stores, centers, workplaces, schools, and hospitals and before traveling.

Regarding the availability of a COVID-19 vaccination center in the workplace, the majority of participants reported that there was a vaccination center near their workplace, and most had already received a second dose of the vaccine. One study found that people are more likely to get vaccinated if vaccines are offered by a government health agency in multiple locations and are free [27]. Another study found that vaccine availability increases people’s acceptance of the vaccine [1]. This finding suggests that governments may seek to provide free COVID-19 vaccines to people in inaccessible locations to increase the number of people vaccinated.

As for the attitudes toward COVID-19 vaccination, our study findings showed that female participants accepted COVID-19 vaccines more than males. Compared to previous studies, one study showed that female participants refused to be vaccinated or had no intention to be vaccinated [28]. Another study also found that males were more likely to have intentions to be vaccinated than females [24]. However, the reason for the high acceptance rate of females in the result of our study may be explained by the fact that our study had more female participants than males. This could be due to the distribution of sample size, where more than 75% of the study population was female. Similarly, the results showed that participants’ attitudes toward COVID-19 vaccination were significantly greater in the western region, as they were the majority (72.25%) among the participants. On the contrary, it was found that participants who lived in the southern region of Saudi Arabia were more willing to be vaccinated than participants who lived in other regions [22]. The results of our study support the findings of other studies [5,17,23,26,28]. This suggests that the population in Saudi Arabia has a positive attitude toward COVID-19 vaccination. This attitude may be due to the Saudi government’s efforts to raise people’s awareness of the vaccine and encourage them to vaccinate. However, the reason for the high acceptance rate in the western region in the result of our study may be explained by the fact that our study had more participants from the western region than other regions.

Our results also suggest a relationship between the level of education and the number of vaccine doses received. We found that holders of master’s and doctoral degrees were most likely to comply with the vaccination requirement, while holders of bachelor’s degrees were less likely to receive the required two vaccine doses. Similarly, one study found that individuals with a university or post-graduate education were more likely to receive the COVID-19 vaccine than those with a high school education [26]. The results of our study suggest that the level of education has a positive influence on the acceptance of vaccination.

Although an association was found between occupations and main sources of information about COVID-19, three occupational groups indicated that social media was the most frequently used source of information; however, administrators were more likely to receive information from the Ministry of Health than other occupational groups. It has become apparent that healthcare administrators must incorporate a large, continuous stream of information into their assessment of an ever-changing situation such as COVID-19. It is, therefore, not surprising that they have better access to information compared to others. This clearly suggested that social media can be considered one of the quickest and most convenient methods to raise awareness about the vaccine and get the message out to the community, especially for college students, staff and academics. Therefore, it is very important for the Ministry of Health to ensure the reliability of the information published on social media. The Ministry of Health should also conduct comprehensive and reliable educational programs through the media to provide accurate information, dispel public misconceptions about vaccines, allay public concerns about vaccination, and increase society’s support for vaccination. This is because inadequate knowledge and misconceptions influenced by social media can reduce the uptake of COVID-19 vaccination. Faculty members were more likely to receive information from colleagues and professional journals than the other two professional groups. This could be due to their involvement in academic activities and the fact that all of them taught health-related courses in their respective institutions.

Participants agreed that most of the factors influenced their decision to vaccinate with COVID-19 vaccines. However, the factor most frequently mentioned by participants that affected their decision to vaccinate was education about the COVID-19 vaccine and its benefits and side effects, followed by the prescription that the vaccine is required for Haj and Umra (pilgrimage) visas. The factor that was mentioned less frequently was people’s preference to obtain natural immunity.

Many previous studies [17,18,20,21,22,23,25,27,28] have indicated that inadequate information about the safety and side effects of the COVID-19 vaccine may reduce population acceptance of vaccination, among many other factors. Therefore, educational programs and increased public awareness of the COVID-19 vaccine and government vaccination regulations could increase the number of vaccinated and achieve the desired goal of increased herd immunity in the population to defeat the pandemic.

The results of this study highlight some strategies that can be used by governments and agencies to overcome vaccine hesitancy for pandemic diseases. These strategies include raising public awareness of the vaccine, offering free vaccination in inaccessible locations when possible, and mandating in-country vaccination with strict monitoring and follow-up policies when necessary. A positive attitude toward vaccination could have a positive effect on the acceptance of vaccination and the educational level of the population. However, this study’s results might not be generalizable to the general population particularly in understanding the factors that might promote vaccine acceptance in a population. This may be due to the fact that the samples were individuals who were highly educated in health matters.

Therefore, governments need to implement comprehensive and reliable education programs through the media to provide the right information, dispel the public’s misconceptions about vaccines, allay the public’s concerns about vaccination, and increase society’s support for vaccination. Inadequate knowledge and misconceptions influenced by social media can reduce vaccine acceptance. Therefore, education programs and increased public awareness of the COVID-19 vaccine and government vaccination requirements can increase the number of vaccinated and achieve the necessary goal of increasing herd immunity in the community to defeat the pandemic. This study focused on academic staff and students in health colleges in Saudi Arabia which are considered a highly educated population as most of the research focuses on the general population. Focusing on such a population may highlight the factors that assist in boosting vaccination acceptance strategies in case of other future pandemics.

This study had some limitations. A self-reported electronic questionnaire was used to collect data, and this may present some bias in the result. Additionally, it was aimed only at students, faculty members, and staff of health institutions mainly in the western region of Saudi Arabia, who may have a high level of awareness of the COVID-19 vaccine as well as convenient access to it, such as the availability of free vaccines at a vaccination center at their universities. Therefore, a study conducted in other locations with a less privileged population might yield different results. Another limitation was that most participants were vaccinated at the time of data collection because vaccines were widely provided by the government and mandated by authorities, especially for all healthcare workers and students, unless exempted by medical opinion. Therefore, we were unable to investigate associations with the receipt of the vaccine. Moreover, at that time, the health authorities had already given enough time to those working in the medical field to obtain the vaccines, which were free and easily accessible. Therefore, we assumed that the people who had received their first dose of the vaccine were somehow reluctant to receive the second dose. It is quite possible that they waited some time to receive the second dose. However, given the recommendations of public health authorities and the demands of the workplace at that time (the period of data collection), we could not relate this delay to the timing factor. In addition, given the relatively small sample, the results of this study cannot be generalized; therefore, it is recommended that a research study be conducted with a larger sample covering a wider geographic area. Finally, we have not investigated the association between attitudes and uptake of the first or second dose of COVID-19 vaccines; therefore, future studies are needed to explore this niche.

## 5. Conclusions

COVID-19 vaccine acceptance in the Kingdom of Saudi Arabia appears to be relatively high. This could be due to the fact that the vaccines are available in the workplace or are prescribed by the relevant authorities. Females were more willing to accept the vaccines than their male counterparts. The coverage was more in the Western region of the Kingdom compared to others. The vaccine coverage was higher in the western region of the kingdom than in others, and higher educational attainment is correlated with compliance. Professional affiliation, e.g., in healthcare, correlates with access to information, and social media appear to be the most important sources of information about COVID-19. Knowledge of the benefits and side effects, as well as fulfillment of religious obligations, such as Haj or Umrah, might encourage people to accept the vaccine COVID-19.

## Figures and Tables

**Table 1 healthcare-11-00999-t001:** Demographic profile of the participants.

Variable	n	%
Nationality		
Saudi	128	73.99
Expatriate	45	26.01
Educational Level		
Secondary school	20	11.56
Diploma degree/Associate College	7	4.05
Bachelor’s degree	52	30.06
Master’s level	30	17.34
Doctorate/PhD	64	36.99
Job		
Student	28	16.18
Administrative job	37	21.39
Faculty member	83	47.98
Technical worker	9	5.20
Security staff	1	0.58
University hospital (Clinician)	15	8.67
Gender		
Male	43	24.86
Female	130	75.14
Age		
20–30	34	19.65
31–40	65	37.57
41–50	49	28.32
51–60	18	10.40
Above 60	7	4.05
Region		
Central Region	9	5.20
Eastern Region	4	2.31
Northern Region	20	11.56
Southern Region	15	8.67
Western Region	125	72.25
Main Source for COVID-19 Information		
Social media	102	58.96
Ministry of Health (MOH)	46	26.59
Professional colleague	16	9.25
Specialty journal/periodicals	9	5.20
Availability of a COVID-19 Vaccination Center at a workplace		
Yes	152	87.86
No	21	12.14
Workplace		
King Abdulaziz University	103	59.54
King Khalid University	15	8.67
Jouf University	20	11.56
Others	35	20.23

**Table 2 healthcare-11-00999-t002:** Prevalence of COVID-19 Infection.

Variable	n	%
Have you been infected with COVID-19?		
Yes	26	15.03
No	147	84.97
Severity of COVID-19 symptoms for infected subjects (n = 26)		
No symptoms	1	3.85
Mild symptoms	21	80.77
Severe symptoms	4	15.38
Do you have any family members who have had COVID-19?		
Yes	86	49.71
No	87	50.29
Have you received a COVID-19 vaccine?		
Yes	154	89.02
No	19	10.98
How many doses did you receive?		
1st dose	53	34.42
2nd dose	101	65.58
Name of the vaccine you received?		
Pfizer	125	81.16
Oxford/AstraZeneca	26	16.89
Other vaccines	3	1.95
Reasons for not receiving the 2nd dose of the vaccine		
I am planning to receive the vaccine.	8	15.09
I have no intentions of receiving the vaccine.	11	20.75
I believe the vaccines are not safe.	16	30.19
Other reasons	18	33.97

**Table 3 healthcare-11-00999-t003:** Summary Statistics of the Factors influencing receiving COVID-19 vaccination (n = 173).

Variable/Items	M (SD)	SEM	SA n (%)	SWA n (%)	NAD n (%)	SWD n (%)	SD n (%)
Fear from the unforeseen future effects of COVID-19 vaccine would decrease tendency to receive it.	3.95 (1.12)	0.09	65 (39.2%)	58 (34.9)	20 (12.0%)	15 (9%)	8 (4.8%)
Mandating vaccine for travel Haj and Umra would increase tendency to receive it.	4.42 (0.97)	0.07	109 (65.7%)	37 (22.3%)	9 (5.4%)	5 (3.0%)	6 (3.6%)
Free vaccines would increase tendency to receive it.	4.36 (0.99)	0.08	102 (61.4%)	43 (25.9%)	8 (4.8%)	7 (4.2%)	6 (3.6%)
Some people prefer natural immunity this may decrease tendency to receive COVID-19 vaccine.	3.57 (1.18)	0.09	39 (23.5%)	60 (36.1%)	35 (21.1%)	18 (10.8%)	14 (8.4%)
Providing education about COVID-19 vaccine and its benefits and side effect would increase tendency to receive it.	4.45 (0.85)	0.06	105 (63.3%)	42 (25.3%)	14 (8.4%)	1 (0.6%)	4 (2.4%)
Mandating COVID-19 vaccine by law would increase tendency to receive it.	4.20 (1.14)	0.09	96 (57.8%)	35 (21.1%)	17 (10.2%)	9 (5.4%)	9 (5.4%)
Seeing family member or friend receiving COVID-19 vaccine would encourage others to receive it if their experience were positive.	4.40 (0.91)	0.07	101 (60.8%)	49 (29.5%)	6 (3.6%)	5 (3.0%)	5 (3.0%)
People with risk factors tend to receive COVID-19 vaccine.	4.06 (0.95)	0.07	66 (39.8%)	60 (36.1%)	28 (16.9%)	9 (5.4%)	3 (1.8%)
Safety and effectiveness of vaccines would encourage people to receive it.	4.39 (0.99)	0.08	107 (64.5%)	37 (22.3%)	10 (6.0%)	6 (3.6%)	6 (3.6%)

SEM = Structural Equation Modeling; SA = Strongly Agree; SWA = Somewhat Agree; NAD = Neither Agree nor Disagree; SWD = Somewhat Disagree; SD = Strongly Disagree.

**Table 4 healthcare-11-00999-t004:** Frequencies and Percentage of the Participants who Received Two Doses of the Vaccine in Relation to Educational Level.

Educational Level	2nd Dose n (%)	χ^2^	*df*	*p*
Secondary school	7 (35.0)	15.33	4	0.004
Diploma	3 (42.9)			
Bachelor’s degree	19 (36.5)			
Master’s level	21 (70.0)			
Doctorate/PhD	51 (79.7)			

**Table 5 healthcare-11-00999-t005:** Frequencies and percentage for the Main Source of COVID-19 Information for selected Job Categories.

Job	Social Media n (%)	MOH n (%)	Professional Colleague n (%)	Specialty Journal/Periodicals n (%)	*p*
Students	18 (64.29)	8 (28.56)	2 (7.14)	0 (0)	0.021
Administrative staff	22 (59.46)	15 (40.54)	0 (0)	0 (0)	
Faculty member	49 (59.03)	14 (16.87)	13 (15.67)	7 (8.43)	

**Table 6 healthcare-11-00999-t006:** Summary Statistics of the Attitude toward COVID-19 vaccination (n = 173).

Variable/Items	SA n (%)	SWA n (%)	NAD n (%)	SWD n (%)	SD n (%)
I feel there is no such virus and therefore no need for receiving vaccination.	3 (1.7%)	1 (0.6%)	22 (12.7%)	26 (15.0%)	121 (69.9%)
I do not trust COVID-19 vaccine benefit.	17 (9.8%)	24 (13.9%)	29 (16.8%)	35 (20.2%)	68 (39.3%)
I am hesitant to receive COVID-19 vaccine because it has serious side effects and might lead to hospitalization or death.	14 (8.1%)	26 (15.0%)	34 (19.7%)	37 (21.4%)	62 (35.8%)
I feel the reasons for promoting COVID-19 vaccine is for financial profit of pharmaceutical companies.	14 (8.1%)	29 (16.8%)	48 (27.7%)	31 (17.9%)	51 (29.5%)
I feel no need for vaccination to people who had corona virus because they are immune.	19 (11.0%)	25 (14.5%)	45 (26.0%)	43 (24.9%)	41 (23.7%)
I am very concerned that I might have COVID-19 virus at the time of vaccination.	18 (10.4%)	38 (22.0%)	33 (19.1%)	47 (27.2%)	37 (21.4%)
I think receiving COVID-19 vaccine will increase the immunity in the community.	104 (60.1%)	34 (19.7%)	18 (10.4%)	11 (6.4%)	6 (3.5%)
I support receiving COVID-19 vaccine because it will enable to remove all restrictions and going back to normal life pattern.	109 (63.0%)	41 (23.7%)	12 (6.9%)	6 (3.5%)	5 (2.9%)
It is important to receive COVID-19 vaccine to control the spread of the disease.	128 (74.0%)	24 (13.9%)	12 (6.9%)	4 (2.3%)	5 (2.9%)
It is important for me to receive COVID-19 vaccine to protect others and myself.	132 (76.3%)	20 (11.6%)	9 (5.2%)	7 (4.0%)	5 (2.9%)

SA = Strongly Agree; SWA = Somewhat Agree; NAD = Neither Agree nor Disagree; SWD = Somewhat Disagree; SD = Strongly Disagree.

## Data Availability

The data that support the findings of this study are available on request from the corresponding author.

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
