# Peer review of "Prevalence, Attitudes, and Factors Influencing Uptake of the COVID-19 Vaccine in Saudi Arabia"

_healthcare, 2023, doi:10.3390/healthcare11070999_

Round 1

Reviewer 1 Report

Thank you for an interesting paper that will contribute tremendously to the body of knowledge in this field.

Methods

You indicate that data collection was performed using a self-report electronic questionnaire; doesn't this introduce some kind of bis? Perhaps you should account for this in the manuscript, even if it under limitations.

Results

You have enrolled on 173 participants in the study and so, e.g., 128 in Saudi Arabia, 125 in Western Region, etc. The 128 and 125 do not add up to 173 though, please kindly explain this. In addition, the sample is too small to make inferences or comparisons among the different regions.   

A higher number of females compared to males enrolled in the study and therefore this cannot infer that "Female participants accepted COVID-19 vaccines more often 278 than males" as stated in the discussion (line 278). Therefore, only the sentence in line 281-283 can explain this difference. 

Author Response

Dear Editor and reviewers,

We thank you for your thoughtful and thorough review and believe your input has been invaluable in improving our manuscript. We discussed your comments and revised them carefully. All comments raised have been addressed below and highlighted in the manuscript. We look forward to hearing from you in due time regarding our submission and any further questions or comments you may have.

Reviewer 2 Report

The paper concerns COVID-19 vaccine acceptance in Saudi Arabia, especially among academic staff, students and workers of health institutions. It is clearly written, however it appears that it is another, rather 'standard' paper concerning factors influencing peoples' decisions of vaccine acceptance.

Despite that the Authors pointed out that the limitation of the study is a very specific group of participants, they present the obtained resuls in the way suggesting that they are reprentative for the whole country, which obviously is not the case. This should be necessarily corrected.

Moreover, the explanation of the fact that a vaccination acceptance rate was higher among women than among men (which does not agree with a result known from the literature) by the greater number of female than male participants is rather strange. Additionally, the grate role of social media seems to be due to the specificity of the participants gruop (acdemic staff and students).

The two sentences in section 4 are almost identical: "This finding suggests that governments may seek to provide free COVID-19 vaccines to people in inaccessible locations to increase the number of people vaccinated. This finding suggests that governments may seek to provide free COVID-19 vaccines to people in inaccessible places to increase the number of vaccinated."

In light of the fact that this paper is another article about COVID-19 acceptance the Authors should justify that it is worth to be published, i.e., they should clearly state what is the main contribution and what it adds to the knowledge contained in the available literature.

Summarizing, in my opinion the paper could be cosidered for a possible publication after major revision.

Author Response

(The authors gave the same response as above.)

Round 2

Reviewer 2 Report

In the current version of the paper the Authors adressed to some extent the comments and suggestions from my report however, the changes made are not convincing. In particular, the higher vaccination acceptance rate among women than among men is still not explained.

Moreover, there are sentences: "This study focused on academic staff and students in health colleges in Saudi Arabia which are considered a highly educated population as most of research focus on general population. Focusing on such population may highlight the factors that assist in boosting vaccination acceptance strategies in case of other future pandemics." They seem to be not convincing, i.e., the conclusions drawn on the basis of so specific population (very well educated persons) may be very difficult to be used in the context of the whole population of the country.

And last but not least, the Authors still didn't explain what is the main and interesting contribution of their paper in comparison to many similar publications.

So, in my opinion the paper still needs a revision.

Author Response

Dear Editor and reviewers,

We thank you for your thoughtful and thorough review and feel that your input has been invaluable in improving our manuscript. We have discussed your comments and reviewed them carefully. All comments have been incorporated below and highlighted in the manuscript using a track change in the main document/manuscript file. We look forward to hearing from you in due course if you have any further questions or comments.
